# Two Low-Level Feature Distributions Based No Reference Image Quality Assessment

**Hao Fu, Guojun Liu \*, Xiaoqin Yang, Lili Wei and Lixia Yang**

School of Mathematics and Statistics, Ningxia University, Yinchuan 750021, China; 18395001758@163.com (H.F.); yxq258351@163.com (X.Y.); liliwei@nxu.edu.cn (L.W.); yanglixia1982@163.com (L.Y.)
\* Correspondence: liugj@nxu.edu.cn

**Abstract:** No reference image quality assessment (NR IQA) aims to develop quantitative measures to automatically and accurately estimate perceptual image quality without any prior information about the reference image. In this paper, we introduce two low-level feature distributions (TLLFD) based method for NR IQA. Different from the deep learning method, the proposed method characterizes image quality with the distributions of low-level features, thus it has few parameters, simple model, high efficiency, and strong robustness. First, the texture change of distorted image is extracted by the weighted histogram of generalized local binary pattern. Second, the Weibull distribution of gradient is extracted to represent the structural change of the distorted image. Furthermore, support vector regression is adopted to model the complex nonlinear relationship between feature space and quality measure. Finally, numerical tests are performed on LIVE, CISQ, MICT, and TID2008 standard databases for five different distortion categories JPEG2000 (JP2K), JPEG, White Noise (WN), Gaussian Blur (GB), and Fast Fading (FF). The experimental results indicate that TLLFD method achieves superior performance and strong generalization for image quality prediction as compared to state-of-the-art full-reference, no reference, and even deep learning IQA methods.

**Keywords:** no reference image quality assessment; low-level feature; generalized local binary pattern; gradient; deep learning



## 1. Introduction

With the rapid development of information and communication technology, end-users continuously propose higher requirements for high-quality experience [1]. The main target of image quality assessment (IQA) is to design an objective image quality evaluation model, which is consistent with human subjective visual perception. Usually IQA methods include two categories: subjective assessment by humans and objective assessment by algorithms designed. Furthermore, objective IQA indices can be classified as full reference (FR), no-reference (NR), and reduced-reference (RR). Especially, NR IQA methods estimate image quality via computer simulation of human vision system (HVS) only from the distorted image, without any information access to the reference images [2,3], therefore NR IQA is more meaningful in image processing and applications.

Nowadays, deep learning has become one of the most attractive fields in the study of artificial intelligence and machine learning. Among them, the convolutional neural network (CNN) is probably one of the most popular, as a special multilayer perception. With the development of deep learning, CNN has not only produced a large number of variant models, but also made great success in various applications, particularly in tasks involving visual, image, and natural language information [4]. Though deep neural networks are powerful in some certain tasks, they have some apparent deficiencies. First, it is well-known that a large amount of training data are usually required in training. Second, deep neural networks, as a complicated black-box model, theoretically, make it so difficult to analyze the deep structure, and powerful computational facilities are usually required in

the training process. More importantly, the learning performance of deep neural networks depends on careful streamline tuning of huge hyper-parameters, seriously. Recently, a series of deep networks and corresponding combination methods have been proposed to improve training speed and theoretical analysis. In [5], Chen et al. proposed an incremental learning system in the form of flat network without the need for deep architecture, named Broad Learning System (BLS). BLS has been demonstrated to have higher learning accuracy with much faster learning speed. In [6], Zhou et al. proposed a multi-grained cascade Forest (gcForest) method. This method generates a deep forest ensemble, with a cascade structure which enables gcForest to do representation learning. The gcForest has much fewer hyper-parameters and quite robust to hyper-parameter settings than deep neural networks.

Deep network directly provides the original image into the learning algorithm, there is no need to extract hand-crafted features in advance. But it often depends on a lot of training data and GPU, and spends a lot of time and experience on training parameters and network structure, such as [7,8]. In fact, deep neural networks are not the best choice for some tasks at the certain condition. By contrast, classical machine learning methods require the design of a set of features to describe the input information, such as scale invariant feature transform (SIFT), local binary patterns (LBP), histogram of oriented gradient (HOG), then input these features into the shallow classifiers, which have simple and efficient recognition effect, such as support vector machine (SVM), random forest (RF), logistic regression (LR).

Good feature description can effectively improve the performance of pattern recognition system. For images, features can be divided into low-level features and high-level semantic features. Low-level features mainly include: color, texture edge, and shape. High-level semantic features need to identify and interpret objects. How to extract effective features is very important in IQA as well as many other vision tasks. Due to the limitation of computer recognition and image understanding technology, it is impossible to accurately analyze and interpret the semantic features of images. Therefore, researchers often adopt some stable low-level features which are easy to extract by machine, so that well reflection of the human visual perception's characteristics and obtaining objective scores for IQA are consistent with subjective evaluation.

## 2. Related Work

The NR IQA method based on low-level features and the methods closely related to our work are introduced in this section. The LBP feature proposed by Ojala et al., in [9–11] is an effective texture local description feature, which encodes the relative intensity values between the central pixel and surrounding pixels, and has been successfully applied in texture classification, face recognition image retrieval, and other fields. Compared with other local texture descriptors, LBP can capture the local texture features simply and quickly. Only in the past decade LBP has been widely used in NR IQA, and become an active research topic. A novel NR IQA method based on structure and luminance information was proposed in [12], which is obtained by extracting LBP to reflect the structural features of the distorted image and extracting the distribution of normalized luminance values to express the brightness characteristics. In [13], Dai et al. used the LBP operator to extract the structural information from the gradient map and the contrast normalization graph respectively. In [14], Yue et al. proposed a fuzzy NR IQA based on the LBP histogram feature. In [15], Zhang et al. used a Gaussian Laplacian filter to decompose the image into multi-scale sub-images, and explored the weighted LBP histogram as the quality-aware feature to input into the support vector regression (SVR) system for obtaining the quality score.

LBP has been extensively and deeply studied in many fields, but the method of feature extraction is sensitive to image noise, and the recognition effect is susceptible to the environment, which makes the local structure description limited. This still needs to be discussed in theory and algorithm.

Wang et al. [16] proposed structural similarity (SSIM) method, which is a milestone in the field of IQA research. The core idea of SSIM is that the change of the image local

structure can effectively reflect the degradation of image quality. The disadvantage of SSIM for severely distorted or unstructured distorted images is inconsistent with subjective evaluation especially. For this problem, Zhang et al. proposed the feature similarity (FSIM) index by using two low-level features of phase consistency and gradient magnitude to replace the statistical features in SSIM [17]. Liu et al. [18] implemented a gradient similarity method by combining gradient features and pixel difference features, which emphasized the gradient can effectively capture structural and contrast changes. In [19], Xue et al. proposed the gradient magnitude similarity deviation (GMSD) method by using the gradient as the feature and the standard deviation instead of the previous mean as the pooling. As it is well-known that the image gradient is sensitive to image distortion and can well characterize the degree of quality degradation of different local structures in the distorted image. Liu et al. [20] considered that gradient direction feature plays an important role in image quality evaluation. They used relative gradient direction and relative gradient amplitude feature to evaluate image quality by using AdaBoosting back propagation neural network.

In [12–15], the authors used LBP for feature extraction and quality evaluation. LBP has good local description, but poor global performance. Since the gradient can reflect the overall structure information of the image, Refs. [18,19] both used different forms of gradient information for quality evaluation. Inspired by the above methods, we propose an NR IQA method based on two low-level feature distributions (TLLFD). The model extracts two types of complementary low-level feature distributions, which not only enhances computational efficiency, but also improves the description and discrimination of the image. The implementation details of TLLFD include three steps: (1) Using the generalized LBP to obtain the difference between the symbol feature and the amplitude feature respectively, then analyzing the histogram of the two features to describe the image texture change; (2) fitting the probability distribution of the gradient amplitude and using the parameters of the distribution to describe the structural changes of the distorted image; (3) the nonlinear regression model is established by SVR. The effectiveness of the method is verified by a large number of contrast experiments with different methods on four standard IQA databases.

## 3. TLLFD for NR IQA

In this section, the local normalization coefficient is first described as image preprocessing, and then the two low-level feature distributions are introduced. Finally, SVR is used as our nonlinear regression model.

### 3.1. Local Normalization

In applications of image processing, given a distorted color image, the color image is transformed to gray scale first.

$$I(i,j) = 0.2989 \times R(i,j) + 0.5870 \times G(i,j) + 0.1140 \times B(i,j) \tag{1}$$

where $R(i,j)$, $G(i,j)$, $B(i,j)$ represent the three color components of the color image respectively. Then, using the same processing model as [21,22], $I(i,j)$ is normalized locally to obtain the mean subtracted contrast normalized (MSCN) coefficient of the image brightness $\hat{I}(i,j)$.

$$\hat{I}(i,j) = \frac{I(i,j) - \mu(i,j)}{\sigma(i,j) + C} \tag{2}$$

where $i$ and $j$ represent the spatial index of the image respectively, $i = 1, 2, \ldots, M$, $j = 1, 2, \ldots, N$, $M \times N$ represents the size of the image.

$$\mu(i,j) = \sum_{k=-K}^{K} \sum_{l=-L}^{L} \omega_{k,l} I_{k,l}(i,j)$$

$$\sigma(i,j) = \sqrt{\sum_{k=-K}^{K} \sum_{l=-L}^{L} \omega_{k,l} (I_{k,l}(i,j) - \mu(i,j))^2} \tag{3}$$

where $\omega = \{\omega_{k,l} | k = -K, \ldots, K, l = -L, \ldots, L\}$, $(2K+1) \times (2L+1)$ is the size of Gaussian window. Take $K = L = 3$, $\mu$ and $\sigma$ are the mean and standard deviation of the local block of the image. $C$ is a normal number to avoid the denominator to take 0 which selected $C = (\alpha L)^2$, $L = 255$. $\alpha$ is a small constant.

Taking the above local normalization method as the image preprocessing step, the normalized result has a good statistical feature change analysis for the distorted image and test image. At the same time, quantifying these changes will make it possible to predict the distortion type affecting the image and its perceived quality.

### 3.2. Low-Level Feature Distribution

3.2.1. Local Binary Pattern

The traditional rotation invariant uniform local binary pattern (LBP) operator [9–11] can be defined as:

$$LBP_{P,R}^{riu2} = \begin{cases} \sum_{i=0}^{P-1} S(g_i - g_c), & U(LBP_{P,R}) \leq 2 \\ P+1, & \text{others} \end{cases} \tag{4}$$

$$U(LBP_{P,R}) = |S(g_{P-1} - g_c) - S(g_0 - g_c)| + \sum_{i=1}^{P-1} |S(g_i - g_c) - S(g_{i-1} - g_c)| \tag{5}$$

$$S(x) = \begin{cases} 1, & x \geq 0 \\ 0, & x < 0 \end{cases} \tag{6}$$

where LBP superscript "riu2" denotes the rotation invariant "uniform" patterns, and $U$ value represents the number of transitions from 0 to 1 or from 1 to 0 with $U$ value less than or equal to 2. $R$ is the neighborhood radius, $g_c$ is the gray value of the central pixel point, $P$ represents the number of neighborhood pixels around the central pixel point $(x_c, y_c)$, and $g_i$ represents the gray value of the neighborhood pixel point $i, i = 0, 1, \ldots, P-1$.

The rotation invariant uniform LBP pattern eventually generates only $P+2$ dimensional texture features, which include $p+1$ uniform pattern and 1 non-uniform pattern. So, the dimension is significantly lower than traditional LBP. In the specific application, the rotation invariant uniform LBP method still has some limitations in scale size and image noise. Different from the traditional LBP, the preprocessing method of formula (2) is used to extract the LBP features of different scales, which is more expressive and discriminative. For solving the sensitivity of LBP to image noise, Guo et al. [23] investigated completed LBP (CLBP). This method analyzes the LBP algorithm from the perspective of local difference sign-magnitude transform. Therefore, the CLBP method will be explored at each position in the texture image to extract texture features, which is defined as [23]:

$$CLBP_{P,R}^{riu2} = \begin{cases} \sum_{i=0}^{P-1} S(g_i - g_c, T), & U(CLBP_{P,R}) \leq 2 \\ P+1, & \text{others} \end{cases} \tag{7}$$

$$S'(x, T) = \begin{cases} 1, & x \geq T \\ 0, & x < T \end{cases} \tag{8}$$

where $T$ is a threshold parameter to be determined adaptively. If the $T$ value is large, CLBP tends to describe the characteristics of image texture which changes dramatically. Conversely, if the $T$ value is small, CLBP tends to describe the details of image texture information. When $T = 0$, namely $CLBP - S_{P,R}^{riu2}$, similar to $LBP_{P,R}^{riu2}$. When $T \neq 0$, namely $CLBP - M_{P,R}^{riu2}$. Here, $T$ in the image is set to $\frac{1}{P} \sum_{i=1}^{P-1} |g_i - g_c|$.

After applying CLBP operator, local $CLBP-S$ and $CLBP-M$ map can be obtained. Then, the global structural features are extracted from $CLBP-S$ and $CLBP-M$ map as the visibility weighted $H_{GTLBP-S}$ and $H_{GTLBP-M}$ histogram, which are presented as following:

$$H_{GTLBP-S_{P,R,T}}(k) = \sum_{i=1}^{M}\sum_{j=1}^{N}|\hat{I}(i,j)|f\left(CLBPS_{P,R}^{riu2}(i,j),k\right)$$
$$H_{GTLBP-M_{P,R,T}}(k) = \sum_{i=1}^{M}\sum_{j=1}^{N}|\hat{I}(i,j)|f\left(CLBPM_{P,R}^{riu2}(i,j),k\right) \tag{9}$$

Here

$$f(x,y) = \begin{cases} 1, x=y \\ 0.\text{others} \end{cases} \tag{10}$$

where $k \in [0,K]$, $K = 9$ is the maximum value of $GTLBP$ model, $M \times N$ denotes the image size, and $\hat{I}(i,j)$ is MSCN coefficients.

Although LBP method is widely used in many fields, it still needs further research and improvement. Some researchers have begun to study multi-feature fusion, which combines LBP with other features more effectively. Generally, features should be complementary to each other for different types of image databases and different fields. Compared with LBP, the gradient has better ability to describe the edge information of the image.

### 3.2.2. Gradient

The edges often appear in the position where the content of target and background changes, and often represent the contours of target in the images. Therefore, image edge extraction plays a key role in the processing of computer vision systems.

Gradient is usually calculated by convolving the image with a linear filter, such as the classic Prewitt [24] and Scharr [25] filters or other filters for specific tasks. The simplest Prewitt filter is used to calculate the gradient. With Prewitt gradient [24] operator, the partial derivatives $G_x(x,y)$ and $G_y(x,y)$ of the distorted image $f(x,y)$ are calculated as follows:

$$G_x(x,y) = \tfrac{1}{3}\begin{bmatrix} 1 & 1 & 1 \\ 0 & 0 & 0 \\ -1 & -1 & -1 \end{bmatrix} * f(x,y)$$
$$G_y(x,y) = \tfrac{1}{3}\begin{bmatrix} 1 & 0 & -1 \\ 1 & 0 & -1 \\ 1 & 0 & -1 \end{bmatrix} * f(x,y) \tag{11}$$

where the symbol "*" denotes a convolution operation and then the gradient magnitude $G(x,y)$ of the image $f(x,y)$ is computed as:

$$G(x,y) = \sqrt{(G_x(x,y))^2 + (G_y(x,y))^2} \tag{12}$$

Statistical information is an effective and robust way to characterize local features. For example, researchers tend to use probability distributions to fit wavelet coefficients, and use the histogram techniques to capture the distribution of features of LBP output, etc. The Weibull probability density function can be written as:

$$p(x) = \frac{\gamma}{\beta}\left(\frac{x}{\beta}\right)^{\gamma-1}\exp\left(-\left(\frac{x}{\beta}\right)^{\gamma}\right) \tag{13}$$

where $x$ is the image gradient magnitude, $\gamma > 0$ is the parameter of shape, and $\beta > 0$ is the parameter of ratio.

Figure 1 shows a reference image in LIVE database and its five distorted images: JPEG2000 (JP2K) compression, JPEG, White Noise (WN), Gauss Blur (GB), and Fast Fading (FF). Figure 2 shows the gradient amplitude distribution of the six images in Figure 1. Among them, the distribution of WN is more uniform and the peak value of FF is the

highest. This is because image quality degradation arises from distortion and the gradient distribution level is also affected by the distortion amount. The scatter plot of Figure 3 shows the Weibull parameter distribution of the six image in Figure 1. In the longitudinal observation, the separation of FF, GB, and WN is obvious. From the horizontal observation, JPEG is distinguished from the reference image clearly. They are different because image with different distortion types may have drastically different parameter. This illustrates that adopting the sensitivity of the shape and proportional parameters of the Weibull distribution to describe different distortion types is effective. It can be observed from Figure 3 that the gradient size of the distorted image follows a two-parameter Weibull distribution, and the human brain response is strongly correlated with the Weibull image for visual perception [26].

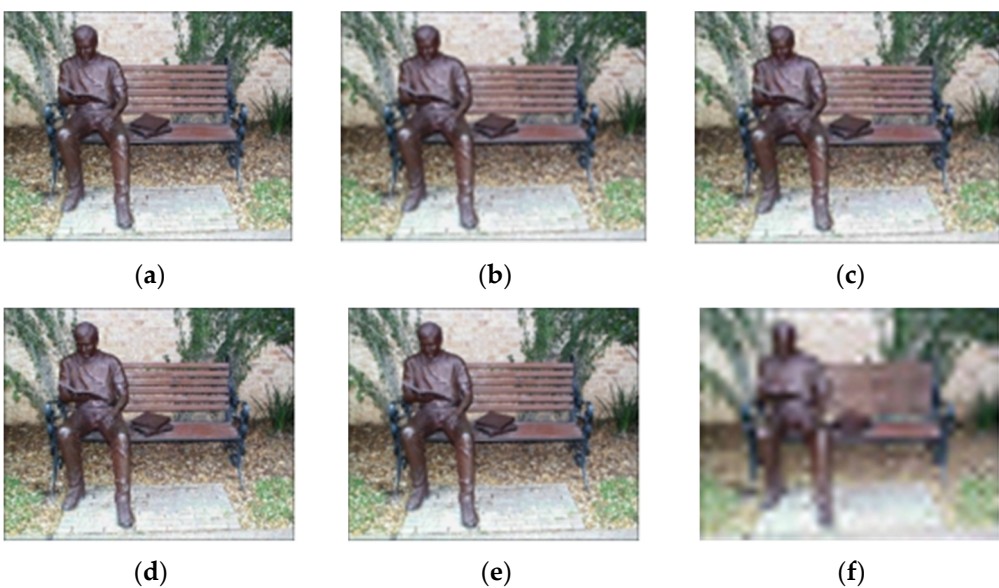

**Figure 1.** A reference image and its five distortion images in LIVE. (**a**) Reference image; (**b**) JPEG 2000 image; (**c**) JPEG image; (**d**) WN image; (**e**) GB image; (**f**) FF image.

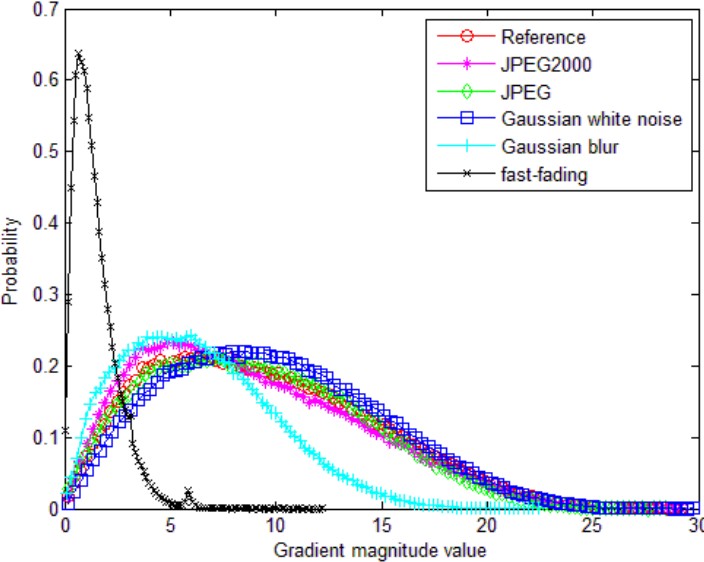

**Figure 2.** Gradient amplitude distribution of different distorted image.

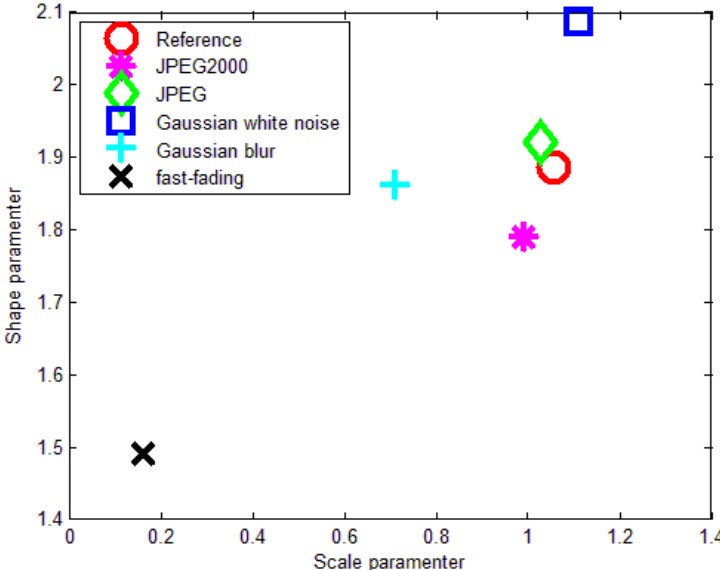

**Figure 3.** Weibull parameter scatter plots of different distorted image.

### 3.2.3. Nonlinear Regression Model

In the next step of feature extraction, the regression function is used to establish the complex nonlinear relationship between feature space and quality evaluation. In [27], Vapnik et al. proposed SVM, which can be regarded as a special type of single hidden layer feed forward network, i.e., support vector network. SVM is a machine learning method based on statistical theory and can transform low-dimensional original feature space into high-dimensional feature space by using kernel function. SVM is generally noted for being able to handle small sample, non-linear, and high-dimensional data. In our implementation, LIBSVM [28] package is used to implement a nonlinear regression model SVR with a radial basis function (RBF) kernel. Considering a set of training data $\{(x_1, y_1), (x_2, y_2), \ldots, (x_l, y_l)\}$, where $x_i \in R^n$ is the extracted quality aware feature and $y_i$ is the corresponding difference mean opinion score (DMOS). Given regularization constant parameters $C$ and constant deviation parameters $\varepsilon$, the standard form of SVR can be represented as [29]:

$$\min_{\omega, b, \xi, \xi^*} \frac{1}{2}\|\omega\|^2 + C\left\{\sum_{i=1}^{l} \xi_i + \sum_{i=1}^{l} \xi_i^*\right\}$$
$$s.t. \begin{cases} \omega^T \phi(x_i) + b - y_i \leq \varepsilon + \xi_i \\ y_i - \omega^T \phi(x_i) - b \leq \varepsilon + \xi_i^* \\ \xi_i \geq 0, \xi_i^* \geq 0, i = 1, 2, \ldots, l. \end{cases} \tag{14}$$

where $\xi_i$ and $\xi_i^*$ are the relaxation variable. $\omega$ and $b$ are the weights and biases, respectively. The parameters $C$ and $\varepsilon > 0$ are found by searching the optimal values from the sets $(2^{-3}, 2^{-2}, \ldots, 2^{10})$ and $(2^{-10}, 2^{-9}, \ldots, 2^6)$.

### 3.3. TllFD Flow Chart and Feature Comparison

In order to further explain the TLLFD method, Figure 4 shows the flow chart.

In this paper, an NR IQA method is proposed. First, the distorted image is downscaled to obtain the scale reduced image; second, the distorted image and the scale down image are locally normalized to obtain the normalized image, and the texture features is extracted from the normalized image. The gradient features are statistically analyzed by Weibull distribution, and two statistical parameters are obtained: $\gamma = 0.2796$ and $\beta = 0.9625$. ($\gamma$ is the parameter of shape, and $\beta$ is the parameter of ratio. In Figure 4, $\gamma$ and $\beta$ only represent). Finally, the extracted features are pooled by SVR to obtain the quality score.

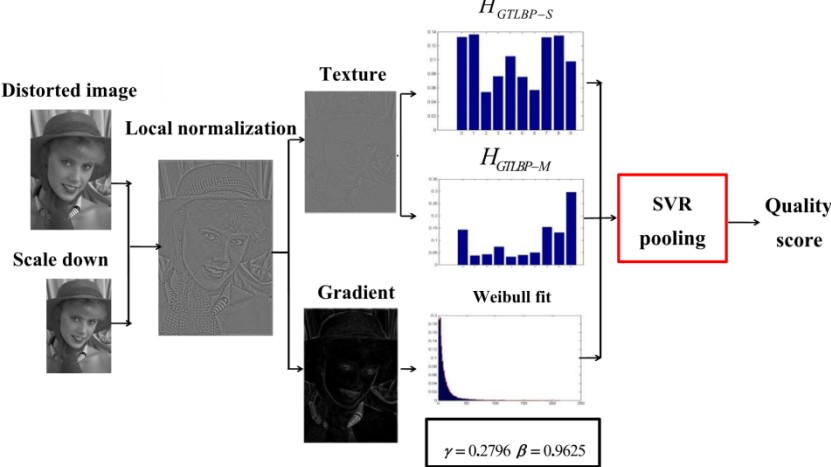

**Figure 4.** Flowchart of TLLFD NR IQA.

In order to further compare the features, this method is analyzed with the four classical methods (NIQE [22], BIQI [30], DIIVINE [31], BRISQUE [21]), as shown in Table 1.

**Table 1.** Feature extraction and regression analysis of different methods.

| Method | Feature Extraction | Regression Method |
|---|---|---|
| NIQE | The normalized information is fitted with the generalized Gaussian distribution to obtain the statistical characteristics | MVG |
| BIQI | Wavelet decomposition is used for feature extraction, and the statistical features are obtained by fitting the generalized Gaussian distribution | SVR |
| DIIVINE | The pyramid wavelet decomposition is used for feature extraction, and the generalized Gaussian distribution is fitted to obtain the statistical features | SVR |
| BRISQUE | (1) MSCN coefficient histogram is fitted with generalized Gaussian distribution; (2) The four direction histograms of MSCN coefficients are distributed by asymmetric generalized Gaussian distribution. | SVR |
| TLLFD | (1) The input image is normalized and preprocessed, extracted CLBP weighted histogram; (2) The gradient features are fitted by Weibull distribution to obtain the statistical features. | SVR |

## 4. Experiments

In this section, the experimental setup is described, including IQA databases, evaluation criteria, and extract feature dimension. Then, TLLFD is compared with classic and state-of-the-art NR IQA models.

### 4.1. Experimental Setups

#### 4.1.1. IQA Databases

TLLFD and state-of-the-art NR IQA models are compared on four standard IQA databases: LIVE [32], TID2008 [33], CSIQ [34], and MICT [35], respectively. The basic

information is listed in Table 2. For CSIQ and TID2008 databases, we only consider the four common distortion types (JP2K, JPEG, WN, GB). In addition, we exclude the 25th synthetic reference image with its distorted versions from TID2008 database.

**Table 2.** Benchmark database for IQA performance validation.

| Database | Source Images | Distortion Types | Distortion Images | Subjects Number | Subjective Scores |
|---|---|---|---|---|---|
| LIVE | 29 | 5 | 779 | 161 | 0–100 |
| TID2008 | 25 | 17 | 1700 | 838 | 0–9 |
| CSIQ | 30 | 6 | 866 | 35 | 0–1 |
| MICT | 14 | 2 | 168 | 16 | 1–5 |

4.1.2. Evaluation Criteria

In order to explain the consistency between objective evaluation and HVS, it is mainly evaluated from the following two aspects: first, accuracy, that is, there is little difference between the results obtained by objective quality evaluation method and subjective judgment. The second is monotonicity. Judging the quality of an image subjectively is consistent with the quality evaluation results obtained by objective methods. The following four evaluation indexes are used for objective quality assessment [36]. Spearman rank order correlation coefficient (SROCC) and Kendall rank order correlation coefficient (KRCC) are used to measure the prediction monotonicity, Pearson linear correlation coefficient (PLCC), and root mean squared error (RMSE) are calculated after the suggested monotonic logistic mapping to measure the prediction accuracy.

For the calculation of PLCC and RMSE, regression analysis is used to provide a non-linear mapping between the objective scores and the subjective mean opinion scores (MOS). For the nonlinear regression, the following mapping function suggested by Sheikh et al. [37] is used.

$$f(x) = \beta_1 \left( \frac{1}{2} - \frac{1}{1 + e^{\beta_2(x - \beta_3)}} + \beta_4 x + \beta_5 \right) \tag{15}$$

where $x$ denotes the original objective score, and $\beta_i(i = 1, 2, 3, 4, 5)$ are regression model parameters to be fitted. A good objective evaluation algorithm has higher SROCC, KROCC, and PLCC, and lower RMSE.

4.1.3. Feature Dimension

For GTLBP calculation, the number of neighbors P is 8 and the radius of the neighborhood R is 1. Different SROCC, KROCC, PLCC, and RMSE indices are obtained by extracting the characteristics of different scales on LIVE. It can be observed from Table 3 that extracting features from two scales is the best, and the performance of our method is relatively stable at different scales. Thus, given a 512 × 512 distorted color image, the extracted features are 44 dimensions in total.

*4.2. Experimental Results and Analysis*

4.2.1. Performance on Individual Databases

In this section, the overall performance of the various IQA models will be tested on each individual database. Each database is divided into training sets and test sets; random selection of 80% of the database constitutes the training set and the remaining 20% makes the test set. Then through 1000 times of cross-validation and the median SROCC and PLCC values are recorded as shown in Table 4. The competing algorithms including four classic ones FR IQA: PSNR, SSIM [16], FSIM [17], VSI [38], eight classic ones NR IQA: NIQE [22], ILNIQE [39], BIQI [30], DIIVINE [31], BLIINDS2 [40], BRISQUE [21], GMLOG [41], NFERM [42], and five state-of-the-art ones deep learning: Dip IQA [43], OG IQA [20], Deep IQA [7], MEON [8], CNN [44].

**Table 3.** Image scale influence on quality assessment.

| IQA Model | Live (799) | | | | |
|---|---|---|---|---|---|
| | SROCC | KROCC | PLCC | RMSE | Dimensions |
| 512 × 512 | 0.949 | 0.811 | 0.951 | 8.358 | 22 |
| 512 × 512<br>256 × 256 | 0.958 | 0.828 | 0.960 | 7.577 | 44 |
| 512 × 512<br>256 × 256<br>128 × 128 | 0.956 | 0.825 | 0.959 | 7.739 | 66 |

**Table 4.** SROCC and PLCC comparison of 15 IQA models on four benchmark databases (the two best models indexes are shown in bold).

| IQA | LIVE (779) | | TID2008 (384) | | CSIQ (600) | | MICT (168) | |
|---|---|---|---|---|---|---|---|---|
| | SROCC | PLCC | SROCC | PLCC | SROCC | PLCC | SROCC | PLCC |
| PSNR | 0.885 | 0.883 | 0.879 | 0.860 | 0.929 | 0.854 | 0.659 | 0.679 |
| SSIM | 0.948 | 0.945 | 0.910 | 0.941 | 0.924 | 0.933 | **0.915** | **0.920** |
| FSIM | **0.963** | **0.960** | **0.953** | 0.929 | 0.924 | 0.912 | 0.906 | 0.800 |
| VSI | 0.952 | 0.948 | 0.940 | 0.923 | **0.942** | 0.928 | 0.866 | 0.736 |
| BIQI | 0.852 | 0.866 | 0.802 | 0.852 | 0.820 | 0.874 | 0.574 | 0.599 |
| DIIVINE | 0.909 | 0.909 | 0.897 | 0.903 | 0.880 | 0.899 | 0.641 | 0.680 |
| BLINDS2 | 0.931 | 0.937 | 0.866 | 0.906 | 0.869 | 0.912 | 0.851 | 0.875 |
| CORNIA | 0.945 | 0.947 | 0.897 | 0.931 | 0.893 | 0.929 | 0.901 | 0.918 |
| BRISQUE | 0.944 | 0.948 | 0.905 | 0.926 | 0.914 | 0.940 | 0.883 | 0.902 |
| GMLOG | 0.950 | 0.954 | 0.937 | **0.945** | 0.923 | 0.951 | 0.885 | 0.888 |
| NFERM | 0.944 | 0.949 | 0.940 | **0.951** | 0.929 | **0.953** | 0.887 | 0.892 |
| OG IQA | 0.951 | 0.955 | 0.937 | 0.941 | 0.924 | 0.946 | - | - |
| CNN | 0.956 | 0.953 | - | - | - | - | - | - |
| Dip IQA | - | - | - | - | 0.930 | 0.949 | - | - |
| Deep IQA | - | - | - | - | 0.871 | 0.891 | - | - |
| MEON | - | - | - | - | 0.932 | 0.944 | - | - |
| TLLFD | **0.958** | **0.960** | **0.940** | 0.944 | **0.939** | **0.953** | **0.919** | **0.925** |

For each criteria, the best two IQA metrics are highlighted in bold. The main observations are as follows. First, TLLFD is closer to the human subjective evaluation of difference mean opinion score (DMOS) on all four databases. Second, TLLFD significantly outperforms PSNR and SSIM. Unfortunately, only CNN evaluation results on LIVE database as well as Dip IQA and Deep IQA evaluation results on CSIQ database are available. Third, compared with other deep learning methods, TLLFD has better quality prediction performance. On LIVE, the SROCC and PLCC values of TLLFD method reach 0.96. On TID2008, the value of SROCC and PLCC are close to 0.94. On MICT, the value of SROCC and PLCC are close to 0.92.

### 4.2.2. Performance on Individual Distortion Types

This section evaluates the performance of NR IQA models on individual distortion types. For NR IQA models, 80% of the five distorted images are used to train the NR IQA model, and 20% of the distorted images with specific distortion types are tested. The SROCC comparison for the 12 NR IQA models in the four benchmark databases is listed in Table 4; the best two NR IQA models for each distortion group are shown in boldface.

In Table 5, we can find that from the results of single distortion type, TLLFD method is better than most methods. For example, according to the experimental results of GB distortion type in CSIQ database, TLLFD method is better than all methods; however, in

other databases, it is not the optimal value, but it is also the suboptimal value. Finally, from the overall weighted average, TLLFD method maintains the optimal value. It should be noted that similar results can be obtained for KROCC, PLCC, and RMSE indicators; only SROCC indicators are listed here. Moreover, the last row of Table 4 lists the weighted average SROCC values of all distortion types, where the weights are the number of images in each distortion group. The quality prediction accuracy of TLLFD is high under individual distortion types on LIVE, TID2008, CSIQ, MICT. For the JPEG2000, JPEG, and FF distortion, it performs slightly worse. For the WN, GB distortion, it outperforms all other NR IQA methods. To sum up, there are the three main reasons for the improvement of performance. First, different distortion types have different CLBP maps, it can effectively measure the influence of different distortion types on image structure change. Second, the global structural feature GTLBP is obtained by weighted histogram, which is an effective descriptor reflecting the effects of different distortion types. Third, JPEG2000, JPEG, FF will cause different degrees of blurring of the image. Blurring reduces the details of image leading to lower performance.

**Table 5.** SROCC comparisons of 12 NR IQA models on individual distortion types (the two best NR IQA models indexes are shown in bold).

| Database | D-TY | NIQE | ILN-IQE | BIQI | DIIV-INE | BLIN-DSII | GM LOG | OG IQA | Dip IQA | Deep IQA | MEON | TLLFD |
|----------|------|------|---------|------|----------|-----------|--------|--------|---------|----------|------|-------|
| LIVE | JP2K | 0.924 | 0.900 | 0.824 | 0.906 | 0.931 | 0.926 | **0.937** | - | - | - | **0.950** |
| | JPEG | 0.942 | 0.944 | 0.884 | 0.897 | 0.950 | 0.963 | **0.964** | - | - | - | **0.962** |
| | WN | 0.972 | 0.979 | 0.965 | 0.982 | 0.946 | 0.983 | **0.987** | - | - | - | **0.987** |
| | GB | 0.940 | 0.924 | 0.856 | 0.934 | 0.915 | 0.920 | **0.961** | - | - | - | **0.958** |
| | FF | 0.862 | 0.844 | 0.743 | 0.854 | 0.875 | 0.901 | 0.899 | - | - | - | **0.907** |
| TID2008 | JP2K | 0.902 | **0.937** | 0.855 | 0.895 | 0.902 | **0.935** | 0.926 | - | - | - | **0.935** |
| | JPEG | 0.887 | 0.887 | 0.887 | 0.887 | 0.887 | 0.884 | **0.934** | - | - | - | **0.931** |
| | WN | 0.817 | 0.883 | 0.756 | 0.840 | 0.685 | 0.891 | **0.907** | - | - | - | **0.904** |
| | GB | 0.847 | 0.860 | **0.899** | 0.890 | 0.857 | 0.886 | 0.881 | - | - | - | **0.929** |
| CSIQ | JP2K | 0.911 | 0.796 | 0.818 | 0.871 | 0.879 | 0.918 | 0.917 | **0.944** | 0.907 | 0.898 | **0.925** |
| | JPEG | 0.913 | 0.828 | 0.859 | 0.883 | 0.895 | 0.917 | 0.933 | 0.936 | 0.929 | **0.948** | **0.942** |
| | WN | 0.925 | 0.924 | 0.850 | 0.901 | 0.868 | 0.946 | 0.941 | 0.904 | 0.933 | **0.951** | **0.949** |
| | GB | 0.883 | 0.905 | 0.844 | 0.895 | 0.883 | 0.915 | 0.907 | **0.932** | 0.890 | 0.918 | **0.936** |
| MICT | JP2K | 0.836 | 0.868 | 0.660 | 0.851 | **0.894** | 0.887 | - | - | - | - | **0.988** |
| | JPEG | 0.906 | 0.868 | 0.690 | 0.755 | 0.873 | **0.941** | - | - | - | - | **0.986** |
| Weighted average | | 0.902 | 0.894 | 0.838 | 0.891 | 0.886 | **0.923** | 0.852 | 0.929 | 0.915 | 0.929 | **0.944** |

### 4.3. Ablation Experiment

4.3.1. Cross-Database Validation and Hypothesis Testing

At the same time, in order to illustrate the generalization capability of TLLFD method and prevent interference from over-fitting experiments, cross database verification experiments are carried out. In order to make a fair comparison, in Table 5, all models are validated on the full LIVE (779) database and tested on TID2008, CSIQ, and MICT. In Table 6, the NR IQA model was trained for image from CSIQ (600) database and tested on three other databases. The two best NR IQA models indexes are shown in bold.

**Table 6.** SROCC comparison on cross-database validation when NR IQA models are trained on LIVE (the two best models indexes are shown in bold).

| Database | NIQE | ILNIQE | BIQI | DIIVINE | BLINDS2 | BRISQUE | GMLOG | NFERM | TLLFD |
|----------|------|--------|------|---------|---------|---------|-------|-------|-------|
| TID2008 | 0.795 | 0.870 | 0.813 | 0.867 | 0.864 | 0.894 | 0.911 | **0.914** | **0.915** |
| MICT | 0.811 | 0.711 | 0.663 | 0.798 | 0.810 | **0.857** | 0.835 | 0.851 | **0.889** |
| CSIQ | 0.869 | 0.880 | 0.785 | 0.877 | 0.902 | 0.890 | 0.899 | **0.907** | **0.928** |

Tables 6 and 7 show the cross database experiment results respectively. Table 6 is the result of taking the LIVE data as the training set and carrying out the experiment in other databases. Table 7 shows the results of experiments with CSIQ data as training set and other databases. From Table 6, we can find that the model trained by LIVE database has better results for other databases, and the TLLFD method is due to the comparison method in the other three databases. From Table 7, we can find that the model trained by CSIQ database has more general results for other databases, but the overall results are good. Except that TLLFD method is slightly worse than nferm method in TID2008 database, it has better results in the three databases when compared with other methods.

**Table 7.** SROCC comparison on cross-database validation when NR IQA models are trained on CSIQ (the two best models indexes are shown in bold).

| Database | NIQE | ILNIQE | BIQI | DIIVINE | BLINDS2 | BRISQUE | GMLOG | NFERM | TLLFD |
|----------|------|--------|------|---------|---------|---------|-------|-------|-------|
| TID2008 | 0.795 | 0.870 | 0.796 | 0.852 | 0.775 | 0.889 | 0.865 | **0.904** | **0.893** |
| MICT | 0.811 | 0.711 | 0.560 | 0.567 | 0.654 | 0.601 | 0.778 | **0.834** | **0.836** |
| LIVE | 0.905 | 0.897 | 0.755 | 0.773 | 0.888 | 0.895 | **0.905** | 0.870 | **0.931** |

Next, to further demonstrate the superiority of TLFFD, we calculated the statistical significance by two sample *t*-tests between SROCC obtained by competing NR IQA methods. The null hypothesis is that the mean correlation of the row is equal to the mean correlation of the column at the 95% confidence level. The alternate hypothesis is that the mean correlation of row is greater than or lesser than the mean correlation of the column.

In Table 8, 1 or −1 indicates that the method is statistically superior or lower than the comparison method, and 0 means has the same effect as the comparison method in statistics. It can be clearly seen from Table 7 that in LIVE database, TID2008 database, and MICT database, the comparison method is shown as 1 in the table, which shows that TLLFD is better than the comparison method. In CSIQ database, TLLFD method has the same experimental results as nferm method, and is better than the other methods.

**Table 8.** Statistical significance *t*-test (1(−1) indicates our method is better (worse) than the method in the column; 0 indicates our method is statistically equivalent to the method in the column).

| *t*-Test | NIQE | ILNIQE | BIQI | DIIVINE | BLINDS2 | BRISQUE | GMLOG | NFERM |
|----------|------|--------|------|---------|---------|---------|-------|-------|
| LIVE | 1 | 1 | 1 | 1 | 1 | 1 | 1 | 1 |
| CSIQ | 1 | 1 | 1 | 1 | 1 | 1 | 1 | 0 |
| TID08 | 1 | 1 | 1 | 1 | 1 | 1 | 1 | 1 |
| MICT | 1 | 1 | 1 | 1 | 1 | 1 | 1 | 1 |

4.3.2. Performance Comparison between LIVEWC and CID2013 Databases

In order to distinguish it from traditional databases, this paper compares the performance of real distortion database and contrast distortion database: LIVEWC, CID2013. The experimental results are shown in Table 9. The results of NR IQA methods with the best evaluation performance are marked in bold. In LIVEWC database, the SROCC value of the proposed method is slightly lower than that of NFERM method, and the PLCC value is higher than that of other NR IQA methods. In CID2013 database, the values of SROCC and PLCC are 0.7786 and 0.7987 respectively, which are obviously superior to other methods. This indicates that TLLFD method has stronger competitiveness compared with other NR IQA methods in LIVEWC and CID2013 databases.

**Table 9.** Performance comparison of two databases with different evaluation algorithms (the two best models indexes are shown in bold).

| IQA Methods | LIVEWC | | CID2013 | |
| --- | --- | --- | --- | --- |
| | **SROCC** | **PLCC** | **SROCC** | **PLCC** |
| BIQI | 0.5324 | 0.5479 | 0.6569 | 0.6757 |
| DIIVINE | 0.5148 | 0.5283 | 0.4972 | 0.5124 |
| BRISQUE | 0.5685 | 0.5864 | 0.4309 | 0.4783 |
| NIQE | 0.4292 | 0.4848 | 0.6007 | 0.6136 |
| BLIINDS2 | 0.4885 | 0.5064 | 0.4766 | 0.4987 |
| NFERM | **0.6055** | **0.5908** | **0.6281** | **0.6322** |
| ILNIQE | 0.5033 | 0.5127 | 0.4540 | 0.4634 |
| TLLFD | **0.6053** | **0.6201** | **0.7786** | **0.7987** |

Two groups of features, texture feature and gradient feature, are extracted from the proposed TLLFD method. In order to explore the contribution of these two groups of features to the final evaluation result, the performance of each group of features on five databases is evaluated respectively. It can be seen from Table 10 that among the five databases, the contribution of texture features is higher than that of gradient features, but the evaluation performance of using one group of features alone is worse than that of using two groups of features simultaneously. This indicates that both sets of characteristics are necessary in the TLLFD method, and they are complementary to the overall evaluation performance.

**Table 10.** Comparison of contributions of the two groups of characteristics (the two best models indexes are shown in bold).

| Database | Evaluation Index | Texture | Gradient | Texture + Gradient |
| --- | --- | --- | --- | --- |
| LIVE | SROCC | 0.9343 | 0.7836 | **0.958** |
| | PLCC | 0.9350 | 0.8063 | **0.960** |
| CSIQ | SROCC | 0.9122 | 0.7230 | **0.939** |
| | PLCC | 0.9213 | 0.7765 | **0.953** |
| TID2008 | SROCC | 0.9263 | 0.7234 | **0.940** |
| | PLCC | 0.9335 | 0.7568 | **0.944** |
| LIVEWC | SROCC | 0.4976 | 0.4346 | **0.6053** |
| | PLCC | 0.5601 | 0.5321 | **0.6201** |
| CID2013 | SROCC | 0.7327 | 0.5234 | **0.7786** |
| | PLCC | 0.7561 | 0.5679 | **0.7987** |

*4.4. Computational Complexity Analysis*

In many practical applications it is desired to estimate the quality of an input image online. Therefore, the computational complexity is also an important factor when evaluating a NR IQA model. The model complexity of the NR IQA model is shown in Figure 5. Our experiments run in Intel Core (TM) i5-3210M CPU @ 2.50 GHz and 4 GB RAM of ASUS A45V laptop. The MATLAB is R2012a (7.14) in the Windows. The 2D scatter plot shows the weighted average SROCC of four standard databases and running time of different methods for feature extraction of a $512 \times 512$ image.

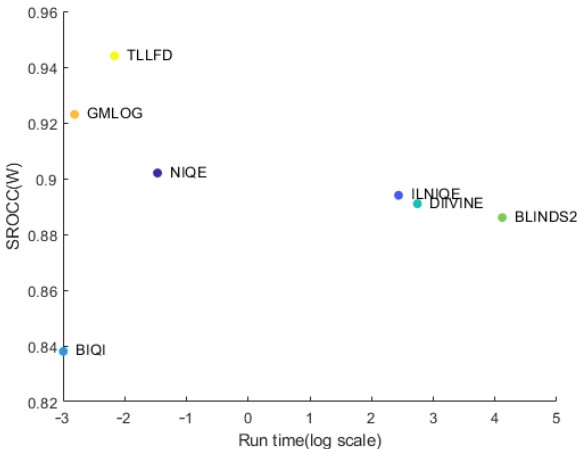

**Figure 5.** SROCC (w) versus running time (log scale) of different methods.

The computational complexity of TLFFD is significantly lower than NIQE, ILNIQE, DIIVINE, and BLINDS2, and worse than BIQI and G-MLOG, which can be easily discerned in Figure 5. The main reasons are as follows. First, BIQI has only two features and extraction time is short. However, its performance is the worst among all the competing models. Second, GMLOG extracts 40 statistical features which combine gradient amplitude and Laplacian features. So, simple extraction process and high operation efficiency are obtained. Third, NIQE has 36 features and a higher computational complexity, leading to less quality prediction performance and slower running speed. Fourth, ILNIQE extracts five types of NSS features and uses them to learn the multivariate Gaussian model and predict the image quality. It has many parameters, which show performance is less competitive. Fifth, the dimension of DIIVINE features is up to 88, thus this model has long running time and low efficiency. Sixth, BLINDS2 using a natural scene statistics model of discrete cosine transform coefficients. The model process is complicated and requires a long running time. Overall, the calculation of TLLFD has low complexity and high efficiency.

## 5. Conclusions

In this paper, we propose a novel framework for NR IQA method, namely TLLFD. First, the normalized information is used as the image preprocessing, which is conducive to the statistical analysis of subsequent features. Then, two low-level feature distributions with unique regression function are extracted, and finally the quality regression analysis is carried out by using nonlinear regression method.

The feature extraction is carried out in two stages. One stage is the weight CLBP histogram coefficients are taken as an image texture feature, and the other stage is the parameters of the Weibull distribution fitting gradient map are used as an image gradien feature.

Verified by the experimental results, the TLLFD method can be compared with the state-of-the-art NR IQA method even when compared with the state-of-the-art FR IQA method. Compared with deep learning-based methods, the TLLFD method also achieves superior performance and strong generalization.

The disadvantage of this method is that some experimental parameters are not adaptively chosen, and knowledge of classical features is needed for deep understanding.

The follow-up work will further study the basic features of the image, and analyze it using the method proposed in this paper. The combination of classical features can have a certain comparative power with the deep learning method, so as to obtain more consistent results with the real MOS.

**Author Contributions:** Conceptualization, H.F., G.L. and X.Y.; Data curation, H.F., G.L., X.Y., L.W. and L.Y.; Formal analysis, G.L.; Funding acquisition, L.Y.; Investigation, X.Y. and L.W.; Methodology, H.F., G.L. and X.Y.; Project administration, L.W. and L.Y.; Resources, G.L. and L.Y.; Supervision, G.L.; Validation, H.F. and L.W.; Writing—original draft, H.F. and L.Y.; Writing—review & editing, G.L. and L.W. All authors have read and agreed to the published version of the manuscript.

**Funding:** This work was supported in part by the National Natural Science Foundation of China (Grant No. 62061040, 12162029, 61941111, 61906102), in part by the Key research and development programs of Ningxia (Grant No. 2019BEG03056), and in part by the Natural Science Foundation of Ningxia (Grant No. 2021AAC03039).

**Institutional Review Board Statement:** Not applicable.

**Informed Consent Statement:** Not applicable.

**Data Availability Statement:** The MATLAB source code of TLLFD is available online at https://github.com/Yazhen1/TLLFD (accessed on 11 May 2022).

**Acknowledgments:** All individuals included in this section have consented to the acknowledgement.

**Conflicts of Interest:** The authors declare no conflict of interest.

## Abbreviations

| | |
|---|---|
| NR | No reference |
| TLLFD | Two low-level feature distributions |
| JP2K | JPEG2000 |
| WN | White Noise |
| GB | Gaussian Blur |
| FF | Fast Fading |
| IQA | Image quality assessment |
| FR | Full Reference |
| RR | Reduced-Reference |
| HVS | Human Vision System |
| CNN | Convolutional Neural Network |
| BLS | Broad Learning System |
| SIFT | Scale Invariant Feature Transform |
| LBP | Local Binary Patterns |
| HOG | Histogram of Oriented Gradient |
| SVM | Support Vector Machine |
| RF | Random Forest |
| LR | Logistic Regression |
| SVR | Support Vector Regression |
| SSIM | Structural SIMilarity |
| FSIM | Feature SIMilarity |
| GMSD | Gradient Magnitude Similarity Deviation |
| CLBP | Completed LBP |
| DMOS | Difference Mean Opinion Score |
| SROCC | Spearman Rank Order Correlation Coefficient |
| KROCC | Kendall Rank Order Correlation Coefficient |
| PLCC | Pearson Linear Correlation Coefficient |
| RMSE | Root mean Squared Error |
| MOS | Mean Opinion Scores |
| NIQE | Natural Image Quality Evaluator |
| ILNIQE | Integrated Local NIQE |
| BIQI | Blind Image Quality Indices |
| DIIVINE | Distortion Identification-based Image Verity and INtegrity Evaluation |
| BLIINDS2 | Blind Image Integrity Notator Using DCT Statistics |
| BRISQUE | Blind/Referenceless Image Spatial QUality Evaluator |
| GMLOG | Gradient Magnitude Map and the Laplacian Of Gaussian |
| NFERM | NR Free Energy-Based Robust Metric |

| | |
|---|---|
| Dip IQA | Discriminable Image Pairs Image quality assessment |
| OG IQA | Oriented Gradients Image Quality Assessment |
| Deep IQA | Deep Image quality assessment |
| MEON | Multi-task End-to-End Optimized Deep Neural Network |
| PSNR | Peak Signal to Noise Ratio |

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
