# Peer review of "Two Low-Level Feature Distributions Based No Reference Image Quality Assessment"

_applsci, doi:10.3390/app12104975_

Round 1

Reviewer 1 Report

The paper is well presented. However, it would be nice if use passive sentence instead of using active sentence. e.g. avoid using "we". In line 78, remove "etc". otherwise state all of them. In line 87, spelling error, "low-lever" should be "low-level feature". In line 136 change "." to ";" . Fig. 1, should put reference if taken from existing work. 

Reviewer 2 Report

The motivation for their work (rapid development of information and communication technology) is clarified and the authors  introduce two low-level feature distributions (TLLFD) based method for NR IQA. They argue the approach is different from the deep learning method, the proposed method characterizes image quality from the distributions of low-level features, thus it has few parameters, simple model, high efficiency and strong robustness of image and then introduce the two low-level feature distributions.

The introduction section is appropriate; the authors clearly present their plan and create an overview of existing literature, however not much comparison with their own research is indicated; summarizing the literature review simply with “Inspired by the above methods, we propose an NR IQA method based on Two 128 Low-level Feature Distributions (TLLFD)”. I suggest a “placement” of the author’s research idea in every existing literature unit or if this is not possible, create a clearer overview of other author’s activities (for example, both authors in [18] and [19] address gradient similarity method, however the approach in our research is …). The proposed metric is also graphically presented in a figure, however not all elements of the figure 1 are addresses, such as the meaning of value beta and gama as well as other acronyms.         

Section 3 includes a large number of (existing) formulas however hardly any references are included. The mathematical approach to describe local normalization coefficients as preprocessing is a poorly explained and lacks insight into why the authors use them for their research. I suggest a graphical presentation of how existing formulas are re-used in the presented research.

The experiment approach in section 4 also lacks clarification: why Spearman rank order correlation coefficient or Kendall rank order correlation coefficient and others…. are used is not clearly explained nor references to a literature unit with such statistical analysis presentation. The hypothesis are also not clearly presented at one place and only randomly mentioned within the text. The research method lacks structure and should be improved.

Empirical research methods (such as an experiment) typically address limitation as well as internal and external validity which is not included in this paper.

The conclusion is too short and does not include all important findings nor does it address the answers to research questions/hypothesis (Hypothesis Testing is unclear as they were not clearly presented).

Other suggestions:

  • Occasional use of non-scientific language: for example: But what kind of statistical distribution does the distorted image gradient magnitude map obey?
  • Names of images in Figure 2 repeat themselves without clarification (only JPEG image is written several times.)
  • Figure 3 a and b should be titled as separate images and references as well as explained as such.
  • The MATLAB source code of TLLFD is available online at

423 https://download.csdn.net/download/weixin36595469/10722188. (the link is not accessible nor  readable for a large part of the potential readers)

Reviewer 3 Report

I propose: 

Move the web link from line 423 to the reference list.

Fig. 4 - please do not use Chinese names.

Please add a list of abbreviations.

Round 2

Reviewer 2 Report

The authors addressed all comments, thank you.